# Experiences and Perceptions of Cervical Cancer Screening Using Self-Sampling among Under-Screened Women in Flanders

**DOI:** 10.3390/healthcare12171704

**Published:** 2024-08-26

**Authors:** Charlotte Buelens, Margot Stabel, Alice Wildiers, Lieve Peremans, Guido Van Hal, Lotje Van Doorsselaere, Annouk Lievens, Alex Vorsters, Severien Van Keer, Veronique Verhoeven

**Affiliations:** 1Department of Family Medicine and Population Health, Primary and Interdisciplinary Care Antwerp, University of Antwerp, 2610 Antwerp, Belgiumlieve.peremans@uantwerpen.be (L.P.); veronique.verhoeven@uantwerpen.be (V.V.); 2Department of Family Medicine and Population Health, University of Antwerp, 2610 Antwerp, Belgium; guido.vanhal@uantwerpen.be; 3Centre for the Evaluation of Vaccination, Vaccine and Infectious Disease Institute, University of Antwerp, 2650 Antwerp, Belgiumalex.vorsters@uantwerpen.be (A.V.); severien.vankeer@uantwerpen.be (S.V.K.); 4Marketing Department, Faculty of Business and Economics, University of Antwerp, 2000 Antwerp, Belgium; annouk.lievens@uantwerpen.be; 5U-MaMi Consortium, 2020 Antwerp, Belgium

**Keywords:** cervical cancer, screening, barriers and facilitators, HPV, self-sampling

## Abstract

Primary Human Papillomavirus (HPV) screening on samples collected by women themselves has proven to be an effective strategy for cervical cancer screening (CCS) and may increase participation rates in women who do not partake (regularly) in screening. The aim of this study is to investigate reasons for non-participation and perceptions of CCS using self-sampling methods among under-screened women in Flanders. Individual interviews with 15 underscreened women aged 30–64 years were conducted. During the interviews, participants were given the opportunity to try out a urine and vaginal self-sampling kit. Thematic analysis was based on Ajzen’s model of the Theory of Planned Behavior. The results showed that 14 out of 15 participants were more likely to participate in CCS if they had the option to self-sample. One of the main findings was a lack of knowledge on different aspects of cervical cancer. Most women had a positive attitude toward cancer screening and primary prevention such as HPV vaccination. Subjective norms were influenced by healthcare professionals, peers, and organized government initiatives. Informational, organizational, contextual, and emotional factors were mentioned as barriers as well as facilitators for participation. Additionally, high self-efficacy was considered to make self-sampling more convenient. All women showed the intention to use at least one method of self-sampling. We can conclude that women who do not participate in CCS would be more motivated to participate using self-sampling methods. Adequate education and guidance should be provided.

## 1. Introduction

Cervical cancer is one of the most common causes of death in women all over the world and is predominantly caused by the human papillomavirus (HPV) [1]. High risk types include HPV 16, 18, 31, and 33 [2]. In the majority of cases, an HPV infection will be asymptomatic and cleared within the first six to 12 months after infection [1]. However, after a persistent HPV infection with high-risk HPV genotypes, the lesion can evolve into cervical intraepithelial neoplasia (CIN), which in turn can progress to cervical cancer. This progress can be influenced by different elements, e.g., smoking, long-term use of oral contraceptives, multiple pregnancies, or pregnancy at a younger age [2]. In 2020, 3381 women in Belgium were diagnosed with in situ cervical cancer and 234 women with an invasive cervical tumor [3].

Flanders has implemented a screening program [4] based on the European Guidelines for Quality Assurance in Cervical Cancer Screening [5], which allows women between the ages 25 and 64 years to be screened for cervical cancer every 3 years by means of a cervical cytological or Papanicolaou (PAP) smear. In this screening program, molecular HPV testing is only performed as a workup of cytological abnormalities. Women are reminded of this screening through the implementation of a call-recall system with an invitation letter. The PAP smear can be executed by a family physician or a gynecologist [6].

Unfortunately, this method does not reach all Flemish women. Data from 2021 indicate that, in Flanders, the total coverage of this screening is 63.6% [3]. This means that one-third of women within the range of 25 to 64 years old were not screened or inadequately screened.

Reasons for non-participation in cervical cancer screening (CCS) programs around Europe are diverse. A suggested alternative in the literature to increase the participation rate is to use self-sampling methods, based on primary HPV screening, which offers higher protection against invasive cervical cancer compared with cytology-based screening [7]. Additionally, self-sampling methods based on HPV-DNA detection by PCR have already been proven to be equally sensitive and slightly less specific than a sample collected by a physician for detecting underlying cervical precancerous anomalies [8]. However, self-sampling methods are currently not used for CCS in Flanders [9]. Over the course of 2024, the recommendations regarding cervical cancer screening will be changed for women aged 30 to 64 years old. Primary HPV screening will be applied in this age group and the screening interval will be extended to once every five years [10].

Overall, previous research in Europe showed a high acceptability of the use of self-sampling kits as an alternative to the PAP smear test [11,12,13]. Self-sampling methods were considered easy to use and overall, women felt confident they completed the sampling properly [9,11,14,15,16,17]. Among the interviewed women, very few experienced pain or discomfort and they registered only a few difficulties in understanding the instructions and in performing the self-sampling [9,11,14,16,17]. Many participants mentioned a preference for self-sampling as compared to physician-collected samples [9,11,14,15,18]. Additionally, women showed a willingness to use the self-sampling methods again in the future and reported that they would participate more often in CCS if it included a self-sampling procedure [11,14]. Furthermore, women would also recommend the use of self-sampling methods to family and friends [9,11]. Fears or worries mentioned in previous research mainly concerned the accuracy of the self-sampling methods and the correct execution of the sample collection [11,13,16,17]. Acceptance of self-sampling is generally high, but differences between women with different backgrounds and literacy levels are apparent [19,20].

Even though extensive research into reasons for non-participation in CCS and the opinions and experiences of women regarding the use of alternative screening methods to increase this screening uptake in Europe has already been conducted, the research with Flemish women as test subjects is rather limited, especially research that delves into women who are harder to reach. Acquiring knowledge to be able to implement the use of these self-sampling methods on a large scale and explore the (cost-) effectiveness is crucial.

Thus, the aim of our study is to explore reasons for non-attendance in the current screening program amongst Flemish women and to become aware of the opinions and experiences regarding the use of urine and vaginal self-sampling kits in this group of non-participating women, which is a population that has not been interviewed sufficiently in Flanders, to ensure that these methods can be applied in future screening programs.

## 2. Materials and Methods

### 2.1. Study Design

This is a descriptive qualitative study using semi-structured individual interviews. For our study, we interviewed Flemish women from the 14 January 2023 until the 26 March 2023.

### 2.2. Study Population and Recruitment

We included 15 women living in Flanders, aged between 30 and 64 years old, who were non-responders in the current organized screening for cervical cancer and who were able to communicate in Flemish. Being a non-responder was defined as not undergoing a PAP smear for at least four years based on self-reporting. Women who had had a PAP smear in the last four years, women with a history of a total hysterectomy or a prior diagnosis of cervical or uterine cancer, or women who were already participating in focus group interviews concerning cervical cancer were not eligible for inclusion. We recruited the participants by using different online platforms (Facebook, Domus Medica, and mailing lists of regional family physician groups) and by contacting different organizations in Flanders that are in close contact with women with a migration background. By using these means of communication for recruitment, we tried to obtain a population with various social and economic backgrounds. Invitation letters were sent out. Additionally, people were asked to participate by social workers and family doctors, a method by which most participants were recruited.

### 2.3. Procedure and Data Collection

Participant recruitment and data collection took place between November 2022 and March 2023. At first, we tried to reach women to participate in focus group interviews. However, despite using the various recruiting methods described above, the initial response was low. Therefore, we decided (with the approval of the Ethics Committee of the UZA/University of Antwerp) to switch to individual interviews. The literature shows that individual interviews as a qualitative research method are as valuable as focus group discussions and can even uncover more information when dealing with sensitive topics (such as, for example, having to take their own vaginal swab) [21].

The semi-structured interviews, conducted in Flemish, took place in Leuven in a university building, in Antwerp at a doctor’s office, and in a local healthcare center. Prior to the interview, women were asked to fill out a demographic questionnaire. During the interviews, all participants were given the opportunity to test two types of self-sampling methods for cervical cancer screening, i.e., Colli-Pee Small Volumes (Novosanis, 2110 Wijnegem, Belgium) and Evalyn Brush (Rovers Medical Devices, 5347 KV Oss, The Netherlands). Informed consent was obtained prior to the interview. The interviews took from 27 to 84 min.

Graphical information on the self-sampling procedures can be found at the following addresses: https://www.dnagenotek.com/US/products/collection-infectious-disease/colli-pee/FV-5000.html and https://www.roversmedicaldevices.com/wp-content/uploads/2023/07/ROVER0194_FLYER_EVALYN_A4.pdf (accessed on 13 August 2024).

### 2.4. Data Analysis

The collected data were stored on external hard drives as audio files, available to all three interviewers. The interviews were transcribed verbatim and coded by three interviewers (CB, MS, and AW). Data analysis started with familiarization of the data and with a primary screening of three of the transcribed interviews, in which we identified four common themes using an open coding approach: incentives and barriers for screening, overall knowledge, and experiences. After this primary analysis, a broader research group (VV, GVH, and LP) discussed the themes and introduced the model of the Theory of Planned Behavior [22]. The original four themes were rearranged into six main themes: attitude, subjective norms, perceived behavioral control, knowledge, intention, and experiences after self-sampling. This framework was used as a guidance for further coding. We coded experiences after self-sampling and knowledge supplementary to Ajzen’s model, considering that knowledge of a particular topic can influence attitude and perceived behavioral control. During the interviews, we explored the participants’ knowledge of a few relevant topics, which included cervical cancer screening, the disease itself, and primary prevention such as vaccination for HPV.

All interviews were coded by at least two of the three interviewers and in the case of uncertainty, they were discussed with the third interviewer. Any changes or additions during the coding of the interviews after finalizing the initial codebook were discussed with all interviewers and, when added to the codebook, were modified in all 15 interviews. Data saturation occurred after 14 interviews, no new information was obtained after this interview. Since our article was written in English, the quotes included in this article were translated from Flemish to English.

## 3. Results

We conducted individual interviews with 15 underscreened women. Their demographics are listed in Table 1. Of the 15 participants, 4 had a migration background. One participant had never had a PAP smear taken before, whereas the others had had at least one PAP smear taken in their lives. Two participants had already taken part in other screening programs in Flanders (i.e., colon and breast cancer screening) and one of the participants had been cured of breast cancer. The average age of our interviewees was 47 years old, ranging from 31 to 62 years old.

Key findings according to the Theory of Planned Behavior are summarized in Figure 1 and detailed in the paragraphs below.

### 3.1. Attitude

All interviewed women had a positive attitude toward cancer screening and thought it was important to detect cancer in an early stage. Some participants had already experienced the importance of screening firsthand because either they themselves or someone they know were diagnosed with cancer.

P4: *“I think, the faster you can detect it, the less damage it can do, so it’s a win-win situation for yourself and for the healthcare system in general. So I think it is very important.”*

P8: *“I am a supporter of it (cancer screening). Like I said, that’s how they were able to diagnose my mother-in-law with colon cancer.”*

Furthermore, all women expressed a positive attitude toward CCS using self-sampling methods.

P11: *“Yes, I would even prefer it, to take a self-sample, because it’s something very intimate. I don’t like to have it done by a gynecologist.”*

Almost all interviewed participants had a positive attitude toward HPV vaccination, which is organized by the Flemish government for boys and girls from 9 to 15 years old. Few participants had been vaccinated themselves but most of them would or already had vaccinated their children (boys as well as girls) against the HPV virus.

P4: *“I don’t believe in all vaccines, but I do believe in this one.”*

P6: *“My daughter has been vaccinated, maybe even 15 years ago, which was even before the vaccination was reimbursed by health insurance, but I thought it was important to do it.”*

Knowledge about the current organized vaccination program varied between participants. One woman had already been vaccinated for HPV but did not know the link between this vaccine and cervical cancer. Incorrect or insufficient knowledge mainly concerned the age and gender of the target population.

P2: *“I also believe that my daughter, who will be 12 in March, that her first vaccination will happen soon, because it’s a couple of times (that they have to get the vaccination) I think, no?”*

P5: *“I know that they use it for young people now but in my time, it wasn’t. But I think it starts at about 16 yo. But when I was young that wasn’t the case yet. I don’t know if that’s still useful now. It could be but if you’re past a certain age, I don’t know…”*

P8: *“I have a little son so it’s not applicable to him.”*

P9: *“I thought they suggested it pretty early, as early as 12–15 yo? I mean, the vaccine, not the screening?”*

### 3.2. Subjective Norms

Participants talked about environmental factors or people that had influenced their motivation to participate in CCS or HPV vaccination programs.

Several women told us that when they had an appointment with their family physician or gynecologist for other reasons (e.g., general checkup, intrauterine device insertion, follow-up for breast cancer, and fertility trajectory), their physician encouraged them to have a PAP smear taken.

P6: *“It is mentioned by the gynecologist on a regular basis. Actually, my family physician who has my medical file has also told me several times that it is time to do it (the PAP smear).”*

Talking about CCS with peers also encouraged women to participate in the program. Someone was convinced by her partner, who worked in a vaccination center, to get their children vaccinated. Someone had lost a family member due to cervical cancer, which was a motivational factor for her to participate in this study.

P3: *“My wife works in a vaccination center, so it would be weird to be an anti-vaxxer. I also remember that before boys were vaccinated (for HPV), she told me she wanted our boys to be vaccinated and she knows a lot more about it so I thought ‘it’s OK’. I also think that it doesn’t do any harm, so I think that every chance to reduce the risk of cancer is good.”*

P8: *“The reason why I participate is because two years ago I lost a cousin because of it (cervical cancer) and I also have a friend who is infertile because of it. So I know it can have far-reaching consequences.”*

In Flanders, the government sends out invitation letters for cervical cancer screening every three years to people who are eligible. Most participants evaluated this as a helpful initiative. However, a lot of women did not remember receiving any of these invitation letters. Even if they did receive an invitation letter, some women would still not participate in the screening program.

P4: *“And then (after receiving an invitation letter) we (participant and her wife) thought it’s been a long time indeed, we need to make an appointment for it, this year we are definitely going to do it (smiling).”*

P11: *“Is it every three years that you get an invitation?” (interviewer: ‘Normally, yes’)*

P11: *“So that’s a reminder, that’s a good thing. It is necessary.”*

### 3.3. Perceived Behavioral Control

The intention to set a certain behavior is also influenced by a person’s perceived behavioral control. Perceived behavioral control can be defined as a person’s perception of the ease or difficulty of performing a given behavior. It is assumed that perceived behavioral control is determined by a person’s overall knowledge and control beliefs, which are beliefs about the presence of factors that may act as facilitators or barriers to performing the behavior.

During our interviews, participants mentioned possible barriers and facilitators for participation in the current CSS program, as well as facilitators and barriers that could influence participation in a possible future screening program using self-sampling kits (i.e., Colli-Pee Small Volumes or Evalyn Brush).

### 3.4. Facilitators

Most of the participants had little knowledge about CCS, the disease itself, and primary prevention. More information and figures about cervical cancer can act as a motivating factor. One participant, a social worker, thought more information about the anatomy of the female reproductive system could motivate people to participate in CCS. Another participant believes that women would possibly be more motivated to participate if it were better highlighted to them that screenings are free.

P2: *“I think it is a good thing that it is for free and that that is clearly stated. I think for a lot of people and for certain communities this is really important.”*

P11: *“And the cervix, is that…. Is that here, is that this cancer? We talk about cervical cancer, but where is that (the cancer)? And then where is the cervix located? Because I also have a uterus, I have two children, but I don’t know… Because YOU (the interviewers) have studied it, YOU know this. But WE don’t. I had absolutely no idea. At first I thought, yes, those are the ovaries, I know that. But then what is this cervical cancer? Is that here or there? In our target group, we always try to work with powerpoints, that they have the information and that they understand it. They’re at a certain age and most of them haven’t even been to school. So you have to try to explain it really low-key.”*

P15: *“I thought it was good to read about the different steps and it is also reassuring, I mean it encourages regular testing because it literally says ‘when you detect early, it’s not such a big problem.’ I think this information in the flyer is really reassuring.”*

#### 3.4.1. Facilitators for Participation in the Current Cervical Cancer Screening Program

None of the participants had had a PAP smear in the last four years even though most of them had a family physician. Improving knowledge about the possibility of having a PAP smear with their family physician might be an organizational threshold-lowering factor because of the trustful relationship.

Motivation to take part in the screening program also depends on certain contextual factors. Some participants had the feeling that if one wanted to have children, CCS suddenly seemed more important to them than before. A couple of participants felt more comfortable doing a PAP smear after giving birth.

P3: *“After giving birth, I didn’t mind that much as before because maybe there was more space…”*

P5: *“If I’m honest, I did it once in my life, that was before I decided to have children, at that moment it suddenly seemed important and before it didn’t.”*

P7: *“I think if I ever have children and then go to a gynecologist, it’s easier because you’re in contact with someone and you know him/her, but for now to go search for someone that I don’t know…”*

#### 3.4.2. Facilitators for Screening Using Self-Sampling Methods

Almost all women described an organizational advantage of self-sampling methods because they are more time-saving and easier to use. You can take the sample wherever and whenever you want. The possibility of self-sampling was also seen as an improvement because of the absence of negative emotions such as shame and discomfort that were linked to PAP smears. With self-sampling, you do not have to expose yourself and you can carry out the sampling in the safe environment of your own bathroom.

P6: *“I think it’s really comfortable (self-sampling) because I don’t like waiting or moving around. Maybe it’s weird but for me it’s a big obstacle to go outside of the house.”*

P8: *“I think the barrier is lower because you don’t have to go to the physician and you don’t have to make an appointment. It can also be an advantage if you feel ashamed to go to the physician for a PAP smear that you can do it yourself.”*

P15: *“For me it’s a good solution because you can do it whenever you want, when you are relaxed. Because otherwise you may be a bit stressed and your body is going to react differently. I would also find it more convenient that I don’t have to go to the physician, and don’t have to make an appointment… I’m not embarrassed about it but I would find it easier that I don’t have to make an appointment because the physician is always busy and I would have to bother him for a PAP smear. However, if something is wrong, I would go, but just for the screening I think it would be OK to do it myself.”*

After reading the instructions that accompanied the self-sampling kits, many women were under the impression that self-sampling would not be difficult, which demonstrated high self-efficacy. However, they did mention that it is important to receive clear instructions and information and that the use of pictograms can improve the clarity of the instructions.

P4: *“I am very curious about the vaginal swab because I think it would solve everything. Because it’s not difficult to do, you can do it at home and then send it or bring it somewhere, it would be really easy.”*

P11: *“A physician is specialized and has studied for it. He is probably better at other types of examinations such as examining the breasts and things like that. But a PAP smear, you can do it equally well yourself.”*

Most participants considered taking a urine sample as the most convenient and less invasive method because you do not have to insert anything into the vagina. Some women compared the vaginal swab with inserting a tampon and found it easy. Many of the participating women had experience with the concept of self-sampling because of the previous use of covid self-tests and considered this a familiar act.

P2: *“I think this (urine sample) is the easiest method and for a lot of women also the least invasive. It’s the same as bringing a urine sample to your physician and a stand to pee prosthetic device is very useful of course.”*

P11: *“It’s the same as a COVID swab, you can do it all by yourself.”*

### 3.5. Barriers

#### 3.5.1. Barriers to Participating in the Current Organized Cervical Cancer Screening Program

Barriers to participating in the current screening program that were mentioned during our interviews can be divided into four categories.

Firstly, participants mentioned different organizational barriers such as not having received an invitation letter. A common remark was the time-consuming nature of making an appointment and going to the hospital or physician’s office. Respondents suggested that employees should be able to book an appointment during office hours or extend the working hours of physician’s practices.

P1: *“It would be good if you have to make the appointment during working hours, you would get paid time off. Because I have two children and I need all my vacation days for their vacation. When you go to the family physician, you can do it in the evening but with the gynecologist it’s not possible. So it would help if it’s possible (to make an appointment) outside the working hours or that you can get paid time off for it.”*

Furthermore, several women did not have a regular family physician or gynecologist and had the impression that the government did not inform them well enough about the topic. One participant mentioned that it is expensive to visit a gynecologist.

P6: *“My gynecologist retired so now I no longer have regular contact anymore. You still have your family physician who knows you, but they know you too well to have your PAP smear taken by them.”*

P7: *“Because the waiting times for appointments are so long, you tell yourself ‘I will do it some other time’ and then you never do it anymore.”*

P13: *“But I don’t seem to have received the invitation, or I might have thought ‘I’ll do it later’ and put it aside and then forgot about it. It must have happened like that, as it does with a lot of people I guess.”*

Secondly, informational barriers were a recurrent theme in our interviews. Half of the participants had hardly any knowledge about the cause, progression, or prognosis of cervical cancer. The most reported risk factor for contracting cervical cancer was penetrative sexual intercourse and some referred to it as a sexually transmitted infection (STI). Some women believed their risk was lower due to the absence of penetration in their specific situations such as a female homosexual relationship, having a partner with prostate cancer, and being a virgin. One woman mentioned that she did not know at which age the incidence of cervical cancer was highest and therefore could not assess if she was at risk or not. An inaccurate assessment of risk factors was made by a couple of women, specifically about a genetic component that had to be triggered.

P2: *“Many people carry it, that’s what I think it was anyway. They do carry the gene, but it doesn’t manifest in everyone, I would say.”*

P3: *“I know it (cervical cancer) is some kind of STI, which is why, because I’m with a woman, my chances are smaller and therefore it’s easier to say ‘meh’. I know men are mainly the carriers, so my risk is smaller but still exists.”*

P5: *“If you know at which age the condition occurs, and you are around this age, you are going to take part in the screening more easily.”*

P6: *“I would say I link it to penetration but that may be completely wrong.”*

Participants lacked knowledge about the current CCS program, namely about screening intervals, the screening age (start and end age), the target group for screening, and the possibility of screening by a family physician. Only a few respondents knew that the screening interval was once every three years and a few did not know they had to repeat a normal PAP smear after three years.

P1: *“Is the screening program not specifically targeted? That’s just for all women?”*

P5: *“I didn’t know you could do that with your family physician, but maybe it’s easier because he’s more of a confidant you see more often than your gynecologist.”*

P6: *“I go to the family physician annually anyway and I think actually a smear test is basically suggested annually, isn’t it?”*

P6: *“No, I think (the screening starts) probably from when you have your period maybe, but could be just as good after pregnancy, I really don’t know.”*

Thirdly, we noticed that specific contexts such as the COVID-19 pandemic could interrupt the regular pattern of repeated PAP smears. Other reasons for non-participation were having no symptoms and previous traumatic experiences with vaginal medication to induce delivery.

P8: *“I used to do it every year, but because of the COVID pandemic, I lost sight of it.”*

P11: *“I can reassure myself, like, there is nothing wrong if I don’t feel bad, so I don’t have to go and do it (PAP smear)”*

Finally, emotions can act as barriers. Participants did not like going to the physician for a PAP smear due to four possible reasons: it is an unpleasant and painful examination, they were nervous about it due to previous traumatic experiences, they found it awkward, or they were ashamed. One participant mentioned that since her most recent follow-up controls for breast cancer were reassuring, she was also more at ease when it comes to cervical cancer.

P2: *“I had a daughter 12 years ago, but my delivery was very difficult. I got induced because I was at risk for preeclampsia because of high blood pressure and they eventually had to insert 36 tablets and then I said ‘now it’s enough, there is only gonna come something out of my vagina, but nothing gets in anymore’. I have booked appointments afterwards, but was always so nervous that I couldn’t go anymore.”*

P7: *“I would consider it and see when I could book an appointment. But if the appointment would then be scheduled after six months, my fear would take the upper hand and I probably wouldn’t do it anymore.”*

P9: *“I had breast cancer, and in the beginning, you get all these therapies and then after a while it’s only hormonal therapy, and then after that only six-month checkups and then yearly and then every two years. In time, it gets less prominent and you feel more secure, and I link the two (cervical and breast cancer), even though I know they are not actually connected.”*

#### 3.5.2. Barriers for the Use of Self-Sampling Methods

Barriers mentioned during the interviews can be divided into three categories. An overview can be found in Table 2.

The first category revolves around information. Too much information on the flyer that accompanies the self-sampling kits was mentioned by multiple participants as a barrier. For one participant, the different steps explained in the vaginal self-sampling kit were not very clear. Women mentioned that they worried language would be a barrier to using self-samples for people with lower educational levels or non-native speakers.

P1: *“It’s a lot of information, however I am used to reading a lot. I don’t know if that’s the case for everyone, but for me it was clear.”*

P2: *“I find it too much to remember to be able to do it in one smooth motion.”*

P4: *“For the classic Flemish inhabitant it will be ok, but for non-native immigrants or people who have difficulties reading, it’s going to be too much. But maybe they don’t go to the gynecologist or family physician either, so you will probably still not reach a part of the population.”*

Secondly, low self-efficacy could act as a barrier. Moreover, knowledge (or the lack thereof) also plays an important role in the evaluation of self-efficacy. Interviewees were familiar with HPV and mentioned that screening consisted of the detection of ‘something’ in the mucosa. However, they could not explain the link with cervical cancer or how one can contract this virus. One participant underwent sequential PAP smears in follow-up after an abnormal PAP smear but could not really tell what exactly was going on at the time, only that her physician mentioned that “they saw something that raised questions and had to be followed up”.

P5: *“Yes, it’s caused by a virus…. Couldn’t it be that maybe you’re already infected, but it just hasn’t come to expression yet? Or is it only because of certain triggers that you develop cancer?”*

P8: *“I know that it can be detected in the lining of the uterus.”*

P9: *“Hmm I know it makes a difference whether it’s HPV positive or negative but the exact link, I can’t really answer that.”*

P15: *“They did say on the phone once briefly something like ‘we’ve seen something,’ I don’t know how they described that, but ‘we’ve seen something that raises a question mark’ or I don’t know how they said that, ‘so we’ll have to do that again after a few weeks’, but actually I never really knew what it was then …”*

Some interviewees were hesitant to use the vaginal self-sampling kit because they had concerns regarding the accuracy of the test if they collected the sample themselves. They believed that the result would be more reliable if the sample was taken by a physician and said it would be important to emphasize that the result of using a self-sampling kit is as reliable as a PAP smear in order to motivate them to take a sample themselves. One of the participants also mentioned she was worried that not everyone would be equally as handy with the self-sampling kits.

P5: *“I think maybe with a gynecologist it will always be better. I would think that I don’t do it well, although with a urine sample you can’t do that much wrong, it has to be first void but other than that…”*

P8: *“Some people might not be so handy with this. For me, I am not young, but not old either, so for me it went quite smoothly.”*

P15: *“I might want to do it, but I would worry that I wouldn’t do it correctly, like with the COVID self-tests, when you didn’t put it deep enough. So I would think ‘did I do it right?’. You always trust someone else who does it for you a bit more.”*

Thirdly, although women mentioned mostly positive emotions when discussing the self-sampling methods, we noticed that fear could act as a possible barrier. Some women mentioned they were a bit frightened by the size of the Evalyn Brush and one participant was worried that when the sample was transported by the mailing service, it would become damaged.

### 3.6. Intention

Intention can be defined as a person’s readiness to perform a certain behavior, and as mentioned above, based on the Theory of Planned Behavior, it stems from a person’s attitude toward the behavior, their perceived behavioral control, and subjective norms. During all 15 interviews, participants showed the intention to test at least one of the two self-sampling kits.

### 3.7. Behavior

During our interviews, participants were given the possibility (without any obligations) to test the Evalyn Brush and Colli-Pee small-volume self-sampling devices if they wanted, to experience self-sampling. The samples were not sent to the lab for analysis.

Of the 15 participating women, 9 tested both the urine and vaginal self-sampling kits, 3 tested only the vaginal, and 2 tested only the urine kit.

Only 1 of our 15 interviewees did not want to test any of the samples, even though she initially did show intention to test the urine kit. After we mentioned that the urine sample would not be sent to the lab for analysis, she found it a waste of material and decided not to take the urine sample. Thus, even though a person has a certain intention (in this case, to take the urine sample), existing barriers (in this case, not obtaining any test results afterward) can still prevent them from actually performing the behavior.

This is something we also noticed when we examined the current screening program for cervical cancer based on PAP smears. During our interviews, a lot of women expressed their intention to participate again in the CCS.

### 3.8. Experiences after Self-Sampling

Most women reported a smooth collection of both samples. They were self-assured that the collection happened correctly. A participant discussed having some traumatic experiences with the delivery of her child but ended up testing the vaginal swab and did not feel like it was traumatic at all because (she stated that) it was much smaller than a speculum.

P1: *“My hands weren’t dirty at all, it turned out much better than I expected. Like it said here, you don’t have to worry, I thought it (the urine sample) was going to overflow… But it is well designed so that the tube isn’t totally full.”*

P15: *“Because I once did it at the family physician, for me this (the vaginal swab) felt the same. I thought maybe I won’t feel anything, but I did feel that something happened. This gave me a reassuring feeling that it would be good.”*

Some participants experienced difficulties with the collection of the vaginal sample. The withdrawal of the pestle of the vaginal swab was especially difficult for them. Other participants found the insertion of the swab inconvenient or were feeling insecure about its quality. Furthermore, after taking the vaginal self-sample, some women experienced a weird feeling in the abdomen accompanied by some blood loss, which made them feel anxious. Only a few participating women had spillages when taking the urine sample.

P1: *“The only thing was that with the vaginal swab, when you have to pull to get the white brush back inside, you have to pull really hard and that didn’t work immediately with me.”*

P3: *“The peeing is a bit messy, you pee a bit on the tube and, when you get up too fast, also on the toilet seat. When it’s at home, it’s no problem but here (the public restroom) you think about the person that comes after you.”*

At the end of the interview, we asked the participants if this method of screening would make it easier for them to participate. Everyone except one participant said that it would make screening more convenient for them and that they thought they would participate more easily in the future.

P4: *“This just comes to your house, it’s not invasive, you can do it whenever you want, it would definitely help. I would definitely participate.”*

P7: *“Once you have started and you go step by step, it was very clear and easy to use. If they would send me this every year, no problem, I would do it every year.”*

The self-sampling method women preferred was variable. A minority of women clearly stated that the vaginal swab was the easiest. Some had a clear preference for the urine sample. The other women had no clear preference.

## 4. Discussion

This study explored barriers to participation in the current organized screening program for cervical cancer, as well as opinions and experiences regarding the use of urine and vaginal self-sampling kits amongst under-screened Flemish women. Barriers to participating in the current screening program were organizational as well as emotional and informational. Previous research shows that having a PAP smear taken is a process that a lot of women find rather daunting due to the fact that it puts them in an uncomfortable or unfamiliar situation. Therefore, the manner in which the caregiver deals with the patient and guides them through the process is of utmost importance [23].

A key finding was the lack of knowledge about cervical cancer in this group of women. Knowledge about screening, as well as the cause, risk factors, progression, and prognosis of cervical cancer was low.

Based on Ajzen’s model of Theory of Planned Behavior, we identified several factors that influence the choice to perform a particular behavior, in this case self-sampling. This group of women had an overall positive attitude toward the use of self-sampling for CCS. Subjective norms influencing screening behavior consisted of peers, healthcare professionals, and the government (by means of the organized screening program invitations). Perceived behavioral control to set a specific behavior is based on facilitators and barriers. The facilitators for self-sampling mentioned during our interviews were mainly practical (e.g., less time-consuming) and emotional (e.g., less shame and discomfort compared to PAP smears). Barriers on the other hand were mostly based on experiencing low self-efficacy, in which a lack of knowledge plays a key role. In this study, all women showed the intention to perform the self-sampling and the vast majority of these women reported a smooth collection of both self-samples and being self-assured about the self-sampling process.

Our findings are in line with previous research about reasons for non-participation in CCS programs around Europe, where emotional barriers (e.g., embarrassment, insecurity, and anxiousness), practical barriers (e.g., accessing appointments), and a lack of knowledge and information were also mentioned [24,25,26]. Furthermore, previous research also shows lower attendance rates in CCS in non-native women, as well as women living in urbanized regions and with a lower socioeconomic status [24,27].

Self-diagnosis has been emerging in the last few years [28]. To overcome the above-mentioned barriers and increase the participation rate in CCS, a variety of interventions and strategies have already been tested, with self-sampling being one of them. Differences in acceptance of the self-sampling methods were seen between different cultures. For example, white women more readily opt for an HPV self-sample, whereas women from an Asian, African, or mixed background tend to opt for a smear test [19]. To improve the health literacy of self-collection for HPV-based screening, it is essential to provide culturally sensitive health education that not only takes into account women’s diverse socioeconomic backgrounds but also literacy levels and lack of self-efficacy [20].

In Flanders, previous research has shown that directly mailing an HPV self-collection kit to non-responders of the regular screening program significantly increases the response rate compared to a recall letter to have a PAP smear taken by a physician [29]. However, in The Netherlands, where the use of self-sampling has been incorporated into the organized CCS since January 2017 and participants can choose between a PAP smear or a self-sampling kit for primary HPV-based screening, a decline in participation rates was seen since the implementation of this strategy [30]. An important side note, however, is that participation in breast and colorectal cancer screening in The Netherlands in recent years has significantly declined as well [31].

Therefore, the use of self-sampling on its own so far does not seem to be sufficient to enhance cervical cancer uptake. The World Economic Forum states the following on its website: “We need multi-stakeholder collaboration to create equitable access to HPV vaccination, screening and treatments” [32]. Engaging and involving stakeholders (e.g., governments, scientists, healthcare professionals, patients and citizens, and providers and industries) is crucial for the dissemination of correct information on CCS and self-sampling. Most importantly, it is crucial to raise awareness of the benefits of screening and avoid social stigma [33,34,35]. This is in line with previous research, which has emphasized the importance of providing education about CCS to both family physicians as well as potential participants (e.g., in educational sessions, group meetings, and leaflets) to enhance participation rates in CCS [36,37,38,39,40]. Additionally, studies demonstrated that different methods of receiving the kit could also influence the participation rate, e.g., home-mailed or family physician-delivered kits [41]. Lastly, walk-in clinics were also proposed as a solution for women who perceived making an appointment as a barrier to participating in CCS [42]. These efforts can be linked to the different factors influencing innovation adoption and diffusion according to Rogers (2003) [43]. Health educational campaigns supported by several stakeholders can make the self-sampling devices, as well as their purpose and impact, more visible to potential adopters, i.e., the nonparticipants in this study. This has been referred to in innovation research as the observability of the innovation. In addition, evidence-based communication by multiple stakeholders may enhance the perceived relative advantage of the CCS self-sampling devices. As indicated, different means of making the self-sampling kits available foster the so-called triability that may positively impact self-sampling adoption. Triability refers to the possibility for potential adopters to test and try the new device kit. In this study, women’s reactions to the self-sampling methods indicated a relatively high degree of compatibility as the use of the self-sampling devices was quite familiar to other testing methods (e.g., COVID-19). A final factor that may trigger innovation adoption that was also demonstrated in our study was the level of complexity since the findings illustrated mixed experiences of respondents when testing the two devices. The latter has been corroborated in research as CCS and self-sampling method adoption may depend on health literacy.

For future research, we, therefore, suggest supporting the use of self-sampling methods by providing more information on cervical cancer and on the reliability of the different self-sampling methods to encourage women to participate in this screening. In this regard, the family physicians, as well as general practice nurses and pharmacists in Flanders, could play an important role. Or, as seen in the GRECOSELF trial [14], midwives could be engaged to provide instructions and guidance before using the self-sampling kit for the first time. Hence, future research will need to address significant collaboration across multiple stakeholders to address disparities in healthcare access and delivery to ensure the effective implementation of primary HPV screening and self-sampling methods.

### Strengths and Limitations

This study presents several strengths. First of all, we were able to include a wide variety of women in the interviews, including hard-to-reach women who have not been interviewed quite as often in previous research. Furthermore, we included women of different ages, sexual preferences, and with different backgrounds (including immigrants). Consequently, we obtained a large variety of opinions and experiences on self-sampling for HPV-based screening. Secondly, only a few studies on self-sampling in CCS have been conducted in Flanders. Therefore, this study adds new insights, especially regarding the applicability of self-sampling in the near future. Thirdly, almost all women took at least one sample, which allowed them to express a reliable opinion about their own experience. Lastly, to prevent interviewer and reporting bias, all interviews were coded by at least two interviewers each time and in case of any doubt, they were discussed with the third interviewer.

There were also some limitations. Women who met our inclusion criteria were part of a hard-to-reach group, which meant we had a rather small number of participants that might not be representative of the general population. However, data saturation was reached after 14 out of 15 interviews. Even though this was not within the scope of our current publication, we do acknowledge that for future research, more quantitative research designs with a larger population, e.g., a survey, would be interesting to obtain more conclusive and generalizable results. Additionally, a significant number of participants were highly educated and a couple of participants mentioned that they had a close relative or friend who was or had been a cancer patient. Therefore, these findings might not be as easily transferable to clinical practice and selection bias cannot be ruled out.

## 5. Conclusions

We conclude that women who do not participate in cervical cancer screening, including hard-to-reach women, would be more motivated to participate if offered the option to do so using self-sampling methods. However, a lack of knowledge is an important barrier to participating. Therefore, adequate education and guidance by an external party should be offered, which could be taken up by the family physician or other healthcare workers.

## Figures and Tables

**Figure 1 healthcare-12-01704-f001:**
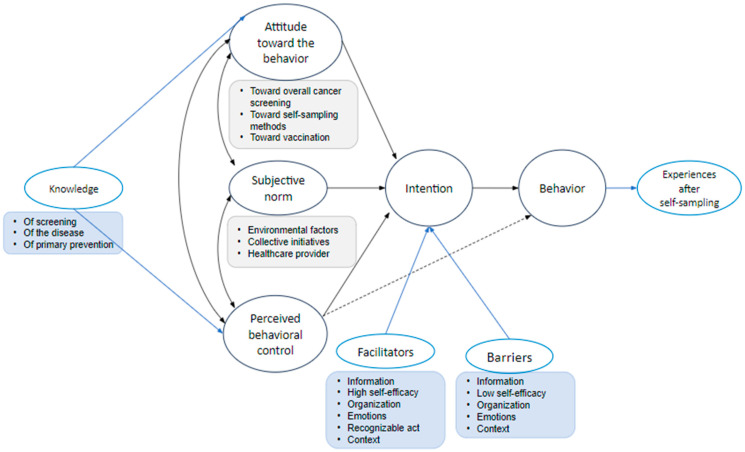
Ajzen’s model of the Theory of Planned Behavior with adaptations. Figure adapted from [22].

**Table 1 healthcare-12-01704-t001:** Participant demographics.

**Age**	** *N* **
30–39 yo	6
40–49 yo	3
50–59 yo	3
60–62 yo	2
Not known	1
**Educational level**	** *N* **
Secondary education	1
Post-secondary education	14
**Has a regular family physician**	** *N* **
Yes	13
No	1
Unknown	1
**Has already consulted a gynecologist**	** *N* **
Yes	14
No	1
**Is aware of the organized cervical cancer screening program**	** *N* **
Yes	13
No	2
**Knows you can have a PAP smear taken with the family physician**	** *N* **
Yes	11
No	4
**Previous experience with PAP smears**	** *N* **
Yes	14
No	1
**Vaccination status for HPV**	** *N* **
Vaccinated	3
Not vaccinated	9
Unknown	3
**Migration background**	** *N* **
Yes	4
No	11

Abbreviations: N, number of participants; yo, years old; PAP smear, Papanicolaou smear; HPV, Human Papillomavirus.

**Table 2 healthcare-12-01704-t002:** Barriers to screening were mentioned during the interviews.

	Mentioned Barriers for Screening in the Current Organized Cervical Cancer Screening	Mentioned Barriers for Screening by Using Self-Sampling Methods
**Organization**	Not having received an invitation letter; Not enough time to make an appointment; Not having a regular family physician/gynecologist; Receiving too little information from the government on the organized screening program and cervical cancer; Finding it expensive to go to the gynecologist.	
**Information**	Lack of knowledge about the cause, progression, prognosis, and risk factors of cervical cancer; Lower perceived risk of cervical cancer; Not knowing at which age incidence of cervical cancer is highest; Lack of knowledge of cervical cancer screening; Not knowing the possibility of having a PAP smear taken at a family physician’s office.	Too much information on flyers accompanying the test kits; Unclear steps in flyers; Language barrier.
**Context**	The COVID-19 pandemic; Not having any symptoms; Previous traumatic experiences.	
**Emotions**	Evaluating the PAP smear as unpleasant, painful, awkward; Experiencing shame, nerves, and fear; Being reassured by breast cancer checkup appointments.	Experiencing fear of the size of the vaginal brush; Fear of the sample becoming damaged in the mailing service;
**Low self-efficacy**		Lack of knowledge of cervical cancer; Lack of knowledge on accuracy of self-samples; Being clumsy.

## Data Availability

The original data generated for this article are available upon reasonable request.

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
