# Peer review of "Experiences and Perceptions of Cervical Cancer Screening Using Self-Sampling among Under-Screened Women in Flanders"

_healthcare, 2024, doi:10.3390/healthcare12171704_

Round 1

Reviewer 1 Report (Previous Reviewer 5)

Comments and Suggestions for Authors

1. At the end of the Introduction, AIM of the study must be also addressed from the perspective of describing the contribution to the field under analysis and the elements of scientific novelty presented, as the LAST, SEPARATE paragraph of this section, to be easier visible. Develop it better. What differentiate your paper from other in the same topic? Give a reason for interest in this paper. 

2. Biomechanism figures must be provided to make it easier for readers to understand this manuscript.

3. How is the sample size determined? What sample size formula do you use?

Author Response

Reviewer 2 Report (Previous Reviewer 3)

Comments and Suggestions for Authors

Questions had been addressed and overall quality of English language had been improved. However, the total number of interviewees in this research is only 15, which greatly affect the scientific soundness of this study. The conclusion may be biased, as the individuals who choose to participate in the interviews might represent a subset of the population that is more motivated to engage in CCS. The authors should conduct the study with a much larger population. Instead of relying on qualitative interview data, researchers should conduct a survey using quantitative methods to obtain more conclusive and generalizable results.

Round 2

Reviewer 2 Report (Previous Reviewer 3)

Comments and Suggestions for Authors

Agree to accept.

This manuscript is a resubmission of an earlier submission. The following is a list of the peer review reports and author responses from that submission.

Round 1

Reviewer 1 Report

Comments and Suggestions for Authors

The writing of the article is not attractive to the reader

It is recommended to revise and rewrite the overall manuscript.

1. The HPV virus is not the only cause of cervical cancer, so it is necessary to address the issues that increase the risk of infection in the introduction.

2. Not all types of HPV cause cancer. Please consider this in your writing.

3. Examining the virus infection is not only by cytological examination of the PAP sample; molecular tests are more helpful for screening, prognosis, and early diagnosis of this disease.

4. Continuity of content in the manuscript is very important. Please review your manuscript based on this comment.

5. It seems that the purpose of the study is not attractively stated. Please revise it.

Comments on the Quality of English Language

 Moderate editing of the English language required

Reviewer 2 Report

Comments and Suggestions for Authors

Attached the review report 

Reviewer 3 Report

Comments and Suggestions for Authors

By interviewing 15 underscreened women, the authors investigated into the reasons for non-participation in cervical cancer screening (CCS) programs implemented in Flanders and how urine and vaginal self-sampling kits could potentially increase this screening uptake among un(der)-screened women. Overall, the manuscript is well-structured and fit the scope of the journal. Just a few minor suggestions to strengthen the manuscript:

1.     Abstract part: please remove works like “BACKGROUND” “METHODS” “RESULTS” “CONCLUSION” and integrate the context into a paragraph.

2.     Line: Please include the total number of interviewed Flemish women in this sentence.

Reviewer 4 Report

Comments and Suggestions for Authors

In their qualitative exploratory study, the authors aimed to explore underscreened women’s reasons for non-adherence to the organized cervical screening program in Flanders as well as their perceptions regarding the use of self-sampling-methods. The study is well conducted and the manuscript well written. I provided a couple of suggestions that could increase the clarity of the manuscript. 

Introduction

Page 2, lines 69-70, the authors may want to revise the statement as it is not clear ” In the course of 2024, the recommendation for cervical cancer changes for women aged 30 to 64 70 years old.

Page 2, lines 73-87, is the provided synthesis is based on studies using qualitative or quantitative methodologies? Does the evidence provided in this paragraph apply to adequately screened or underscreened individuals (or both)?

Page 3, line 96, one of the objectives was to explore women’s opinions about urine and cervico-vaginal self-sampling kits. The authors could mention that while HPV testing on cervico-vaginal samples is recognized as being effective, not enough data has accumulated yet about the sensitivity and specificity of HPV testing using urine samples and to my knowledge none of the countries that have transitioned to HPV testing in primary cervical screening has incorporated in their programs urine-based testing.

Materials and methods

It would be useful to provide details about the development and content of the interview topic guide. Was it informed by the TPB? Could the authors provide the interview guide as supplementary material?

The authors used the TPB for deductive thematic analysis. Were any themes generated inductively?

Results

Table 1, the authors provide the category “60 and older” but the age range was 31 to 62 years. Therefore, for consistency, this category should be renamed to “60 to 62 years”

The sample (Table 1) does not appear to reflect highly vulnerable women as most had good education attainment, had a regular family physician, had a history of Pap screening and were not migrants. Therefore, the conclusions of this study may not be transferable to women experiencing a poor socioeconomic status or significant health inequities. The authors might want to acknowledge this as a study limitation.  

I was surprised to see attitudes (as exemplified by quotes) about HPV vaccination because exploring women’s knowledge, attitudes and beliefs about HPV vaccination was not among the stated objectives of the study. Did the interviewers specifically ask questions about HPV vaccination? On the same vein, it is surprising that only one statement about self-sampling is included in the attitudes category; did the interview topic guide include any question about self-sampling?

The facilitators and barriers refer to the “perceived benefits” and “perceived barriers” of the Health Belief Model (HBM). Planned Behavioral Control integrates barriers and facilitators into a single perception of control, whereas HBM treats them as separate influences on the decision-making process. From the synthesis/description provided in the manuscript I am inclined to believe that in most cases these barriers and facilitators influence women’s decision-making than their perception of control. Therefore, I am unsure if these categories (facilitators and barriers) provided in Figure 1 influence behavioral control or if in fact they influence intentions and behaviors i.e., the arrows should point toward “Intention” and Behavior”.

Comments on the Quality of English Language

N/A

Reviewer 5 Report

Comments and Suggestions for Authors

Determining the sample size must be based on statistical analysis, so before carrying out this study, statistical analysis to determine the sample size must be carried out, because this greatly affects the validity of a study result.

Round 2

Reviewer 1 Report

Comments and Suggestions for Authors

Thanks to the authors for revising the manuscript and improving the quality of the article. 

Please check quality of figures.  

Reviewer 5 Report

Comments and Suggestions for Authors

Authors failed to respond to reviewer comments in the previous review phase.
